

# Chemical patterns of colony membership and mother-offspring similarity in Antarctic fur seals are reproducible

Jonas Tebbe[1], Emily Humble[1,2,3], Martin Adam Stoffel[1,4], Lisa Johanna Tewes[5], Caroline Müller[5], Jaume Forcada[3], Barbara Caspers[6] and Joseph Ivan Hoffman[1,3]

[1] Department of Animal Behaviour, Bielefeld University, Bielefeld, Germany
[2] Royal (Dick) School of Veterinary Studies and the Roslin Institute, University of Edinburgh, Edinburgh, UK
[3] British Antarctic Survey, Cambridge, UK
[4] Institute of Evolutionary Biology, University of Edinburgh, Edinburgh, UK
[5] Department of Chemical Ecology, Bielefeld University, Bielefeld, Germany
[6] Department of Behavioural Ecology, Bielefeld University, Bielefeld, Germany

Corresponding author
Joseph Ivan Hoffman,
joseph.hoffman@uni-bielefeld.de

## ABSTRACT

Replication studies are essential for evaluating the validity of previous research findings. However, it has proven challenging to reproduce the results of ecological and evolutionary studies, partly because of the complexity and lability of many of the phenomena being investigated, but also due to small sample sizes, low statistical power and publication bias. Additionally, replication is often considered too difficult in field settings where many factors are beyond the investigator's control and where spatial and temporal dependencies may be strong. We investigated the feasibility of reproducing original research findings in the field of chemical ecology by performing an exact replication of a previous study of Antarctic fur seals (*Arctocephalus gazella*). In the original study, skin swabs from 41 mother-offspring pairs from two adjacent breeding colonies on Bird Island, South Georgia, were analyzed using gas chromatography-mass spectrometry. Seals from the two colonies differed significantly in their chemical fingerprints, suggesting that colony membership may be chemically encoded, and mothers were also chemically similar to their pups, hinting at the possible involvement of phenotype matching in mother-offspring recognition. In the current study, we generated and analyzed chemical data from a non-overlapping sample of 50 mother-offspring pairs from the same two colonies 5 years later. The original results were corroborated in both hypothesis testing and estimation contexts, with *p*-values remaining highly significant and effect sizes, standardized between studies by bootstrapping the chemical data over individuals, being of comparable magnitude. However, exact replication studies are only capable of showing whether a given effect can be replicated in a specific setting. We therefore investigated whether chemical signatures are colony-specific in general by expanding the geographic coverage of our study to include pups from a total of six colonies around Bird Island. We detected significant chemical differences in all but a handful of pairwise comparisons between colonies. This finding adds weight to our original conclusion that colony membership is chemically encoded, and suggests that chemical patterns of colony membership not only persist over time but can also be generalized over space.

Our study systematically confirms and extends our previous findings, while also implying more broadly that spatial and temporal heterogeneity need not necessarily negate the reproduction and generalization of ecological research findings.

## INTRODUCTION

Replication studies are fundamental to the scientific process as they are essential for evaluating the correctness of scientific claims and the conclusions of other scientists (*Schmidt, 2009*). Indeed, *Fisher (1974)* recommended that a null hypothesis should always be rejected more than once because "no isolated experiment, however significant in itself, can suffice for the experimental demonstration of any natural phenomenon". Nevertheless, replication studies are still "troublingly rare", particularly in fields such as ecology and evolutionary biology (*Nakagawa & Parker, 2015*). *Palmer (2000)* argued that we ignore reproducibility at our peril because this perpetuates a "contract of error" that undermines our understanding of important ecological and evolutionary phenomena.

There has also been debate and confusion over exactly what constitutes reproducible research (*Mendoza & Garcia, 2017*). *Goodman, Fanelli & Ioannidis (2016)* recognized three basic concepts, (i) "methods reproducibility", which requires that the methodology of a given study be provided in sufficient detail to allow it to be repeated; (ii) "results reproducibility", often known as "replication", which is the ability to corroborate previous results using the same experimental methods in a new study; and (iii) "inferential reproducibility", which relates to whether or not qualitatively similar conclusions are reached on the basis of either an independent replication of a study or a re-analysis of the original data. Furthermore, replication studies can be "exact", meaning that they show a high degree of fidelity to the original experiment, "partial", which involves procedural or methodological changes, or "conceptual", where the same questions are investigated but using different approaches (*Kelly, 2006*). The latter two categories include "quasi-replication" studies, which extend the scope of the original study beyond the specific system or species in question (*Palmer, 2000*). In general, the closer the replication attempt is to the original study, the more valuable are the results for assessing the validity of the original claims (*Nakagawa & Parker, 2015*). However, quasi and conceptual replications are also important because they can shed light on the generality (also known as "transportability") of the effects under investigation (*Goodman, Fanelli & Ioannidis, 2016*; *Dirnagl, 2019*; *Piper et al., 2019*; although see *Kelly, 2006*). Put another way, it is only possible to learn something about the broader significance of a certain effect by probing to what extent it persists in settings that are different from, or which lie outside of the experimental framework of the original study. Quasi and conceptual replications therefore play an important role in increasing the "external validity" of results (*Schmidt, 2009*).

Another conceptual difficulty relates to the basis on which replication success is judged. Although there is no single standard for evaluating replication outcomes, most replication attempts are deemed successful if a null hypothesis that was rejected in the original study is again rejected (*Rosenthal, 1991*; *Kelly, 2006*). However, due to the dependance of *p*-values on sample sizes, success or failure in attaining significance may not always provide a good measure of replication success (*Kelly, 2006*). Consequently, several authors have advocated reporting effect sizes and associated measures of precision, as these allow replication outcomes to be gauged in a continuous manner rather than on the basis of binary significance outcomes (*Kelly, 2006*; *Goodman, Fanelli & Ioannidis, 2016*; *Piper et al., 2019*).

In recent years, high-profile failures to reproduce a significant proportion of studies in the medical and social sciences (*Begley & Ellis, 2012*; *Open Science Collaboration, 2015*, reviewed by *Kelly (2019)*) have led to a crisis of confidence (*Baker, 2016*). The generally poor success of replication studies has been attributed to a "publish or perish" culture that incentivizes dubious research practices such as selectively reporting significant results, *p*-value hacking and establishing hypotheses after the results of a study are known (*Fidler et al., 2017*; *Fraser et al., 2018*). All of these practices increase the risk of false positives and contribute towards publication bias (*Jennions & Møller, 2002*), which undermines the robustness of the scientific literature. Further issues include poor study design, low statistical power, variability in reagents or the use of specialized techniques that are difficult to repeat, lack of scientific oversight, inadequate reporting of data, methods and results, and insufficient incentives for sharing data and code (*Baker, 2016*; *Fidler et al., 2017*; *Piper et al., 2019*).

Despite growing awareness of these issues not being specific to any particular scientific field, ecological and evolutionary studies are seldom replicated, with only around 0.02% of studies having been self-reported as exact replications (*Kelly, 2019*). One reason for this may be the general perception that research in these fields can be difficult to replicate, partly due to the complexity and lability of many of the phenomena under investigation, but also because in many field situations replication may be unfeasible or even unethical (*Kelly, 2006*; *Nakagawa & Parker, 2015*; *Fidler et al., 2017*). Furthermore, numerous factors cannot be controlled for in natural settings and environmental variation in particular may confound attempts to reproduce previous results (*Kelly, 2006*). However, these are not valid reasons to neglect replication studies as it is important to understand the extent to which research outcomes hinge upon these and other factors.

The field of chemical ecology provides an interesting case in point. Increasing numbers of studies are using approaches like gas chromatography-mass spectrometry (GC-MS) to characterize the chemical composition of biological samples such as skin swabs or urine. The resulting "chemical fingerprints", otherwise commonly referred to as "chemical profiles", "scent profiles" or "odour profiles" (*Hurst & Beynon, 2010*), comprise multiple peaks that are separated according to their retention times and which represent different substances. Studies of both captive and wild animal populations have shown that these chemical fingerprints can convey information about species identity (*Caspers et al., 2009*; *Fratini et al., 2012*; *Krause et al., 2014*), population membership (*Schneeberger et al., 2016*;

*Wierucka et al., 2019*), sex, age and reproductive state (*Caspers et al., 2011*; *Kean, Müller & Chadwick, 2011*; *Vogt et al., 2016*), family membership (*Sun & Müller-Schwarze, 1998*; *Müller & Müller, 2016*), individual identity (*Kean, Chadwick & Mueller, 2015*; *Kohlwey et al., 2016*), social status (*Burgener et al., 2009*) and genotype (*Yamazaki et al., 1990*; *Charpentier, Boulet & Drea, 2008*; *Setchell et al., 2011*). However, concerns have been raised over the small sample sizes of many studies, which afford little statistical power and may ultimately lead to effect sizes being overestimated (*Wyatt, 2015*). Furthermore, GC-MS data are inherently noisy, making peak detection and alignment challenging (*Ottensmann et al., 2018*). The failure to report peak detection and alignment methods in sufficient detail might therefore act as a barrier to the successful replication of chemical studies. Finally, chemical fingerprints are complex and multidimensional, being influenced by a multitude of factors (*Hurst & Beynon, 2010*; *Stoffel et al., 2015*) including both intrinsic (e.g., genes, hormones and metabolic status) and extrinsic (e.g., environmental variation and diet) variables. Consequently, it remains unclear to what extent many chemical patterns will be repeatable, particularly under natural and often highly heterogeneous conditions.

Pinnipeds provide interesting model systems for studying chemical communication as they possess large repertoires of functional olfactory receptor genes (*Kishida et al., 2007*) and are sensitive to even the faintest of smells (*Kowalewsky et al., 2006*). Many pinnipeds have a strong musky smell (*Hamilton, 1956*), which has been attributed to facial glands that hypertrophy during the breeding season (*Ling, 1974*; *Hardy et al., 1991*), suggesting an important role of olfactory communication during the peak reproductive period. Olfaction may be particularly crucial for mother-offspring recognition because females of many pinniped species accept or reject pups after naso-nasal inspection (*Kovacs, 1995*; *Dobson & Jouventin, 2003*; *Phillips, 2003*). Indeed, a study of Australian sea lions showed that mothers are capable of discriminating their own pups from nonfilial conspecifics based on odor alone (*Pitcher et al., 2011*). This discovery motivated our team to perform a study of Antarctic fur seals (*Arctocephalus gazella*), in which chemical fingerprints were characterized from skin swabs taken from 41 mother-offspring pairs at two breeding colonies—the special study beach (SSB) and freshwater beach (FWB)—at Bird Island, South Georgia (*Stoffel et al., 2015*). Despite being separated by less than 200 m, animals from these two colonies exhibited highly significant chemical differences, while mothers also showed greater chemical similarity to their pups than expected by chance.

Although further research is needed, these findings may have implications for the social organization of Antarctic fur seals as well as for individual recognition. On the one hand, chemical differences between animals from different colonies could potentially facilitate colony recognition and thereby help to explain the remarkable natal philopatry and breeding site fidelity of this species (*Hoffman, Trathan & Amos, 2006*; *Hoffman & Forcada, 2012*). As a result, it is possible or even likely that chemical communication will influence the local relatedness structure of fur seal breeding colonies with downstream impacts on inbreeding and mate choice (*Hoffman et al., 2007*; *Humble et al., 2020*). On the other hand, chemical similarities between mothers and their pups are consistent with the hypothesis that mother-offspring recognition in this species may involve self-referent

phenotype matching, a conceptually simple mechanism whereby an individual's own phenotype is used as a template for the recognition of close relatives (*Blaustein, 1983*).

Here, we attempted to replicate the chemical patterns of colony membership and mother-offspring similarity reported by *Stoffel et al. (2015)*. We returned to the same two breeding colonies 5 years later, collecting and analyzing chemical samples from 50 new mother-offspring pairs using virtually identical methodology. Because these two studies were carried out several years apart, none of the individuals overlapped, precluding analysis of the reproducibility of chemical patterns within individuals. Instead, we use the term "reproducibility" to refer to the extent to which broad chemical patterns, that is, differences between colonies and similarities between mothers and their offspring, can be replicated with non-overlapping samples from different time points.

In addition, we wanted to know whether chemical differences between animals from SSB and FWB are specific to this particular setting, or whether chemical signatures are colony-specific in general. We therefore analyzed chemical samples from an additional 60 pups from four other colonies around Bird Island in order to test for the generality of the colony membership pattern, by which we mean the extent to which chemical differences are more generally present among animals from different colonies. We hypothesized that (i) the originally reported patterns of colony membership and mother-offspring similarity would be repeatable; and (ii) that animals from different breeding colonies would differ chemically from one another in general.

## MATERIALS AND METHODS

### Study site and fieldwork

Chemical samples were taken from six Antarctic fur seal breeding colonies on Bird Island, South Georgia (54°00′S, 38°02′W) during the peak of the 2016 breeding season (November–December; the previous study was conducted during the peak of the 2011 breeding season). A total of 50 mother-offspring pairs (including one pair of twins) were sampled from SSB and FWB as part of annual routine procedures of the long-term monitoring and survey program of the British Antarctic Survey (BAS). Additional samples were collected from a total of 60 pups from four colonies (15 samples each from Johnson Cove, Main Bay, Landing Beach and Natural Arch, Fig. 1). Here, pups were opportunistically sampled from areas of the beach that were easily accessible. Adult females and pups were captured and restrained on land using standard methodology (*Gentry & Holt, 1982*). Chemical samples were obtained by rubbing the cheek underneath the eye and behind the snout with sterile cotton wool swabs, which were stored individually at −20 °C in glass vials containing approximately 10 mL of 60%/40% (vol/vol) ethanol/water. All of the chemical samples were collected immediately after capture by the same team of experienced field scientists. The samples were frozen at the latest 1 h after collection and were stored for approximately 18 months prior to analysis.

### GC-MS profiling and data alignment

We first took two mL of each sample and allowed the ethanol to evaporate at room temperature for a maximum of 12 h before resuspending in two mL dichloromethane

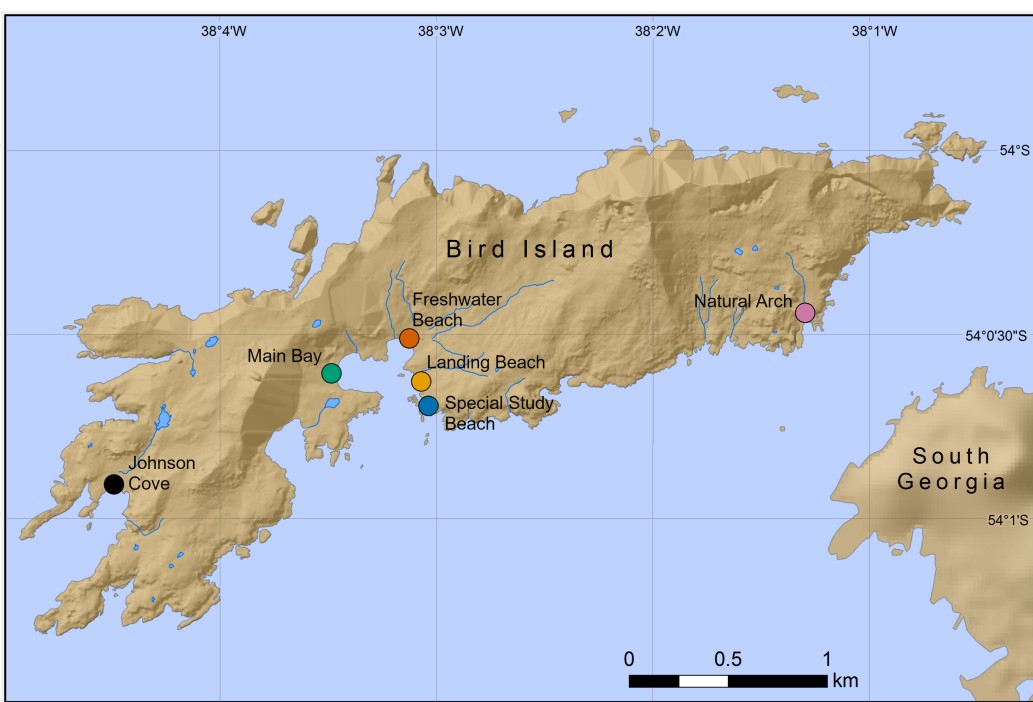

**Figure 1 Locations of six Antarctic fur seal breeding colonies on Bird Island, South Georgia, where chemical samples were taken.** Mother-offspring pairs were sampled from the special study beach (SSB) and freshwater beach (FWB), whereas only pups were sampled from the other four colonies.

(DCM). After a further evaporation step, in which the DCM was reduced to a final volume of approximately 100 μL, the samples were analyzed on a GC with a VF5-MS column (30 m × 0.25 mm inner diameter, 10 m guard column; Agilent Technologies, Santa Clara, CA, USA) connected to a mass spectrometer (GCMS-QP2020, Shimadzu, Kyoto, Japan). One μL of each sample was injected into a deactivated glass-wool-packed liner with an inlet temperature of 225 °C. A split ratio of 3.2 was used and the carrier gas (Helium) flow rate was held constant at 1.2 mL/min. The GC run started with 3 min at 60 °C and then ramped up in increments of 10 °C/min to reach a final temperature of 280 °C, which was maintained for 30 min. Mass spectra were taken in electron ionization mode with five scans per second in full scan mode (50–600 m/z). The resulting GC-MS data were then processed using OpenChrom (*Wenig & Odermatt, 2010*) for detection and correction of split peaks. Afterwards, we used GCalignR (*Ottensmann et al., 2018*; *R Core Team, 2019*) to align the resulting chromatograms by correcting minor shifts in retention times among samples and maximizing the number of shared components.

## Data visualization and statistical analysis

Prior to data analyses, we excluded any compounds that were only observed in a single sample. We then used non-metric multidimensional scaling (NMDS) to visualize the chemical data. This approach reduces dimensionality so that each individual data point can be placed in a 2D scatterplot where ranked between-individual distances are preserved and individuals that are chemically more similar are closer together. NMDS was

performed on a log($x$+1) transformed relative abundance matrix comprising pairwise Bray–Curtis similarity values. We tested for differences among and between a priori defined groups (i.e., the breeding colonies and mother-offspring pairs) using a non-parametric permutational multivariate analysis of variance (PERMANOVA). PERMANOVA tests whether the centroids of pre-defined groups differ statistically for a chosen distance measure. It compares within-group to among-group variance components and assigns statistical significance based on random permutations of objects within groups. Each PERMANOVA was based on 99,999 permutations, although comparable results were also obtained with 9,999, 999 and 99 permutations. To determine whether differences between our pre-defined groups were attributable to compositional differences between groups rather than compositional differences within groups, we used the "betadisper" function in the vegan package in R to analyze the multivariate homogeneity of group dispersions (*Oksanen et al., 2019*). In addition, we performed pairwise PERMANOVAs for different groups within the model strata based on age and colony and Bonferroni corrected the resulting *p*-values.

## Quantification of the explained variance

To facilitate a comparison of our effect sizes with those reported by *Stoffel et al. (2015)*, we quantified the proportion of the total chemical variance attributable to colony membership and family ID in both studies. The scripts that *Stoffel et al. (2015)* used to align their data have now been embedded into GCalignR and are therefore consistent between the two studies. As different chemical datasets will have different optimal parameter settings for the alignment algorithm, we did not re-align or adjust the dataframe of *Stoffel et al. (2015)*. Enforcing the same parameter settings as in the current study would almost certainly lead to a loss of data quality and result in artificially reduced effect sizes. To standardize effect size estimates between the studies, both chemical datasets were bootstrapped over individuals to generate 5,000 datasets per study, each comprising 15 mother-offspring pairs from SSB and 15 pairs from FWB (i.e., a total of 60 individuals). PERMANOVA was then implemented separately for each dataset and the resulting $R^2$ values were extracted for each of the predefined groups.

## Data availability

The raw chemical data generated during this study are available via GitHub and the data of *Stoffel et al. (2015)* can be downloaded from https://github.com/mastoffel/seal_chemical_fingerprints. All of the code used to analyze the raw data are available as a PDF file written in Rmarkdown (see Supplemental Information). The full documented data analysis pipeline can be downloaded from our GitHub repository at https://github.com/tebbej/SealScent2020/.

## Ethical statement

Samples were collected as part of the Polar Science for Planet Earth program of the British Antarctic Survey under the authorization of the Senior Executive and the Environment Officers of the Government of South Georgia and the South Sandwich Islands

(permit no. 2016/013). Samples were collected and retained under Scientific Research Permits for the British Antarctic Survey field activities on South Georgia, and in accordance with the Convention on International Trade in Endangered Species of Wild Fauna and Flora. All field procedures were approved by the British Antarctic Survey Animal Welfare and Ethics Review Body (reference no. PEA6).

## RESULTS

In order to investigate the reproducibility of chemical patterns of colony membership and mother-offspring similarity in Antarctic fur seals, we analyzed chemical data from mother-offspring pairs from SSB and FWB as well as pups from an additional four breeding colonies around Bird Island (Fig. 1). We detected an average of 42 ± 15 s.d. chemicals per sample. No significant differences were found in the number of chemicals between mothers and offspring (unpaired $t$-test, $t = 0.8403$, $p = 0.403$) or among pups from the six breeding colonies (ANOVA, $F_{5,104} = 0.001$, $p = 0.98$).

### Reproducibility of chemical patterns

Multivariate statistical analysis of the relative proportions of each substance revealed highly significant differences between animals from SSB and FWB (PERMANOVA, $p < 0.0001$, Fig. 2A; Table 1A). A highly significant effect of mother-pup pair ID nested within colony (PERMANOVA, $p < 0.0001$, Fig. 2B; Table 1A) was also found, indicating that mothers and their pups are chemically more similar to one another than expected by chance. A test for multivariate homogeneity of group variances uncovered marginally significant differences among the groups ($p = 0.026$, Table 1A), which could potentially indicate the involvement of additional explanatory factors that were not accounted for in the model. We therefore investigated further by splitting the chemical data into four groups, corresponding to mothers and pups from SSB and FWB respectively. Performing PERMANOVAs for all possible pairwise combinations of these groups resulted in three important outcomes. First, all of the pairwise PERMANOVAs involving groups of animals from the two different colonies were highly significant after table-wide Bonferroni correction for multiple tests (Table S1). This indicates that colony membership is chemically encoded irrespective of whether individuals are mothers or pups. Second, both of the pairwise PERMANOVAs involving mothers and pups within colonies were non-significant after Bonferroni correction (Table S1). This suggests that mothers and their pups are chemically similar to one another, regardless of the colony in question. Finally, tests for the homogeneity of group variances were not significant for any of the pairwise group comparisons after Bonferroni correction (Table S2). This implies that our results are unlikely to be driven by differences in chemical variance among groups.

As $p$-values cannot be directly compared between studies with different sample sizes, we used the PERMANOVA framework to estimate the effect sizes of colony membership and mother-offspring similarity in both studies. To facilitate direct comparisons while also incorporating uncertainty due to chemical variation among individuals, both datasets were bootstrapped over individuals as described in the Materials and Methods.

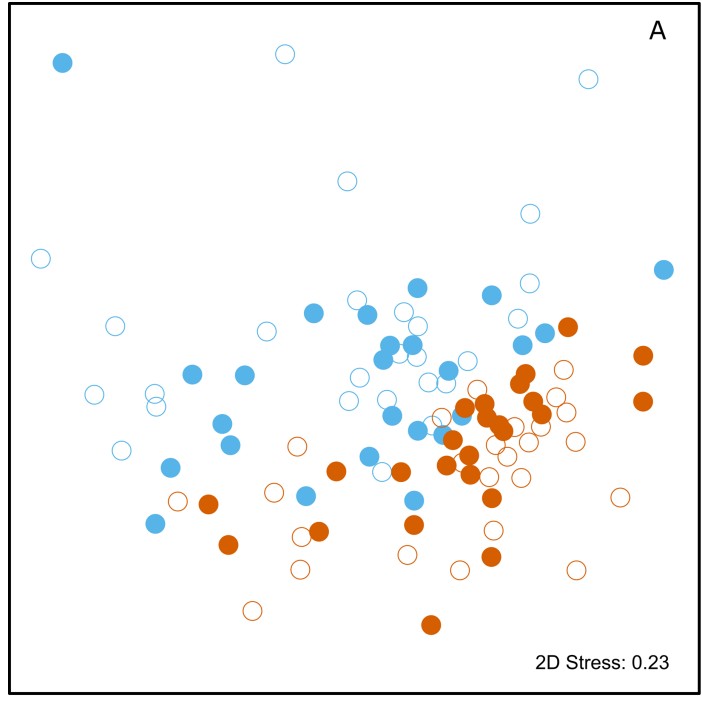

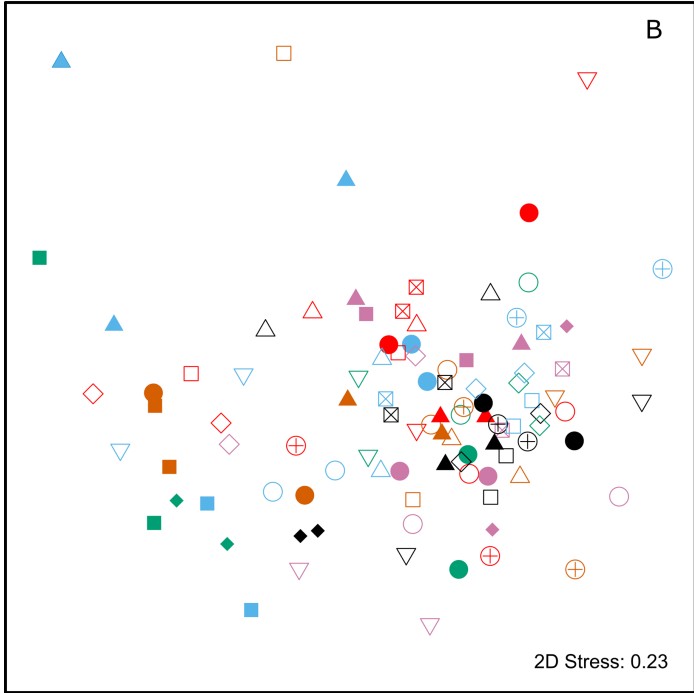

**Figure 2 Two-dimensional non-metric multidimensional scaling (NMDS) plots of chemical data from skin swabs of Antarctic fur seal mother-offspring pairs from SSB and FWB.** NDMS was performed using Bray–Curtis similarity values calculated from log($x$+1) transformed chemical abundance data. The scales of the two axes are arbitrary and the closer two points appear in the plot, the more similar they are chemically. Individual data points in (A) are color-coded by colony (SSB = blue, FWB = red) and age (mother = filled, offspring = empty). (B) Shows mother offspring pairs, which are depicted by unique combinations of symbols and colors. The three blue triangles correspond to a mother on SSB with twin pups.

**Table 1 Results of PERMANOVAs.**

| | $F$ | $R^2$ | $p$-Value |
|---|---|---|---|
| (A) PERMANOVA of mothers and offspring from two colonies | | | |
| Age | 2.65 | 0.023 | 0.004 |
| Colony membership | 9.07 | 0.076 | <0.0001 |
| Family ID nested within colony membership | 9.02 | 0.153 | <0.0001 |
| Test for homogeneity of variance for colony membership | 5.14 | | 0.026 |
| Test for homogeneity of variance for age | 1.47 | | 0.228 |
| Test for homogeneity of variance for age & colony membership | 1.91 | | 0.134 |
| (B) PERMANOVA of pups from six colonies | | | |
| Colony membership | 5.17 | 0.191 | <0.0001 |
| Test for homogeneity of variance for colony membership | 0.50 | | 0.778 |
| (C) PERMANOVA of mothers and offspring from two colonies (data from *Stoffel et al. (2015)*) | | | |
| Age | 0.98 | 0.010 | 0.461 |
| Colony membership | 12.35 | 0.128 | <0.0001 |
| Family ID nested within colony membership | 3.13 | 0.065 | <0.0001 |
| Test for homogeneity of variance for colony membership | 0.22 | | 0.639 |
| Test for homogeneity of variance for age | 0.35 | | 0.557 |
| Test for homogeneity of variance for age & colony membership | 0.21 | | 0.887 |

**Note:**
Results are shown for (A) 50 mother-offspring pairs from two colonies (SSB and FWB); and (B) 110 pups from six colonies. For comparison, a re-analysis of the chemical data of *Stoffel et al. (2015)* is shown in part (C). See "Materials and Methods" for details.

We found that effect size estimates for colony membership and mother-offspring similarity (maximum density $R^2$ values) differed by only few percent between the two studies (Fig. 3) and consistently fell within the range of $0.08 < R^2 < 0.15$.

### Generality of chemical patterns

To investigate whether chemical signatures are colony-specific in general, we analyzed chemical data from pups sampled from a total of six colonies around Bird Island. PERMANOVA uncovered chemical differences not only between SSB and FWB, but also more generally among colonies (Fig. S1). These differences were statistically significant both overall ($p < 0.0001$, Table 1B) and for the majority of pairwise comparisons after Bonferroni correction (Table 2).

## DISCUSSION

A major obstacle to reproducible research in ecology and evolution is the perceived difficulty of replicating original research findings in natural settings where many variables cannot be controlled for and where spatial and temporal dependencies may confound faithful replication attempts (*Nakagawa & Parker, 2015*; *Fidler et al., 2017*). Although the inherent variability of natural systems undoubtedly poses a challenge to replication studies, our findings suggest that, at least under some circumstances, chemical patterns may be repeatable. Specifically, we found that the effect sizes of patterns of colony

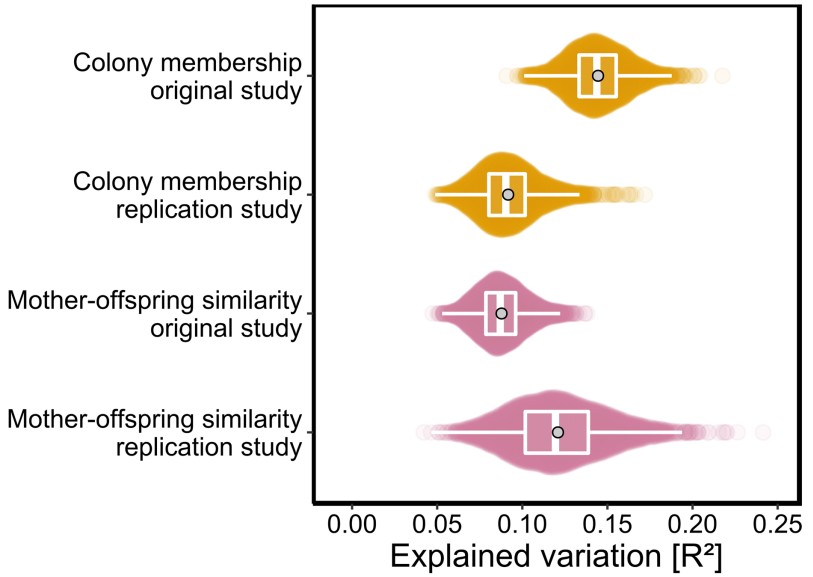

**Figure 3 Effect sizes of colony membership and mother-offspring similarity in the original study (*Stoffel et al., 2015*) and in this replication study.** To quantify the amount of explained variance, we bootstrapped both datasets over individuals and extracted the corresponding $R^2$ values for each of the predefined groups in separate PERMANOVAs (see Materials and Methods for details). The data are presented as sinaplots with overlaid boxplots (centre line = median, bounds of box = 25th and 75th percentiles, upper and lower whiskers = largest and lowest value but no further than 1.5 * inter-quartile range from the hinge) and the gray points represent effect sizes based on the full datasets.

**Table 2 Results of PERMANOVAs of colony membership in pups from six colonies.**

| Pairs | $F$ | $R^2$ | $p$-value | Corrected $p$-value |
|---|---|---|---|---|
| SSB versus FWB | 6.08 | 0.110 | <0.0001 | <0.001 |
| SSB versus Landing Beach | 3.54 | 0.083 | <0.001 | 0.012 |
| SSB versus Main Bay | 6.18 | 0.137 | <0.0001 | <0.0001 |
| SSB versus Natural Arch | 4.17 | 0.097 | <0.001 | 0.002 |
| SSB versus Johnson Cove | 4.40 | 0.104 | <0.0001 | <0.001 |
| FWB versus Landing Beach | 4.16 | 0.099 | <0.0001 | <0.001 |
| FWB versus Main Bay | 7.29 | 0.161 | <0.0001 | <0.001 |
| FWB versus Natural Arch | 7.91 | 0.172 | <0.0001 | <0.001 |
| FWB versus Johnson Cove | 7.15 | 0.162 | <0.0001 | <0.001 |
| Landing Beach versus Main Bay | 3.32 | 0.106 | 0.002 | 0.030 |
| Landing Beach versus Natural Arch | 3.00 | 0.097 | 0.004 | 0.063 |
| Landing Beach versus Johnson Cove | 2.70 | 0.091 | 0.012 | 0.177 |
| Main Bay versus Natural Arch | 5.92 | 0.175 | <0.0001 | <0.001 |
| Main Bay versus Johnson Cove | 3.14 | 0.104 | <0.001 | 0.005 |
| Natural Arch versus Johnson Cove | 2.43 | 0.083 | 0.016 | 0.245 |

**Note:**
The magnitude and significance of chemical differences are shown for all possible pairwise combinations of pups from six breeding colonies. See "Materials and Methods" for details.

membership and mother-offspring similarity in *Stoffel et al. (2015)* were of similar magnitude in a new sample of mother-offspring pairs separated by 5 years. By expanding the geographical scope of our sampling, we could furthermore show that chemical signatures are colony-specific in general. Our results lend further support to the conclusion that colony membership and mother-offspring similarity are chemically encoded in Antarctic fur seals.

## Motivation and study design

A number of factors motivated the current replication attempt. First, the results of *Stoffel et al. (2015)* were based on a modest sample of Antarctic fur seal mother-offspring pairs sampled in a single season. We therefore wanted to safeguard against type I error while also testing for the repeatability of chemical patterns over time. Second, chance results can become highly influential (*Kelly, 2006*) and our original study already appears to have motivated comparable investigations in other pinniped species. For example, a recent study of Australian sea lions using a very similar experimental design also reported chemical differences between two breeding colonies, but chemical similarities were not found between mothers and their pups (*Wierucka et al., 2019*). Although it is not unreasonable to assume that different species might vary in how chemical information is encoded and used in mother-offspring recognition, this point of difference nevertheless encouraged us to revisit our original findings. Finally, being able to confirm and extend our original results strengthens the case for follow-up studies and reduces the risk of time and resources being wasted on chasing up false positives.

Although we acknowledge that no study of a wild population can ever be perfectly replicated (*Nakagawa & Parker, 2015*; *Fidler et al., 2017*), we believe that our replication study of chemical patterns in Antarctic fur seals is sufficiently close to that of *Stoffel et al. (2015)* in terms of both experimental design and implementation to be considered an exact replication. In practice, there were a handful of small differences between the two studies, but these were mainly a consequence of incremental improvements to our methodology and are unlikely to have had a major influence on the final outcome. For example, because replication studies often produce smaller effect sizes than original studies (*Simonsohn, 2015*; *Open Science Collaboration, 2015*), we attempted to enlarge our sample size of mother-offspring pairs as far as was practicable. We also improved the standardization and reproducibility of our chemical analysis pipeline by performing peak detection with open source software and by integrating the alignment algorithm of *Stoffel et al. (2015)* into an R package (*Ottensmann et al., 2018*). However, these small modifications appear to have been of little consequence as the effect sizes of colony membership and mother-offspring similarity did not differ systematically between the two studies.

Two further methodological differences were beyond our control. First, owing to the fact that the original and replication studies were carried out 5 years apart, the sampling was conducted by different teams of field biologists. However, we used carefully standardized field protocols in order to minimize any inadvertent experimental variation. Second, the GC-MS machine used by *Stoffel et al. (2015)* was subsequently replaced

by a newer and more sensitive model. One might have expected this to result in more chemicals being detected in the replication study, which would be expected to provide greater power to detect chemical patterns. If anything, however, fewer chemicals in total were detected in the current study, possibly because of differences in the concentrations of samples or because we used different peak calling software and manually curated the resulting dataset to remove redundant split peaks. Regardless of the exact explanation, the overall similarity of the results of the two studies suggests that patterns of colony membership and mother-offspring similarity in Antarctic fur seals are robust to these minor sources of experimental variation. This robustness would be expected if chemical patterns are influenced by large numbers of compounds and therefore persist independently of minor methodological differences that may influence which subsets of peaks are detected and retained for analysis.

## Replication outcomes

Successful replication can be defined either in the context of statistical significance (*Rosenthal, 1991*) or on the basis of a comparison of effect sizes (*Goodman, Fanelli & Ioannidis, 2016*; *Piper et al., 2019*). We not only tested for significance but also developed an approach based on PERMANOVA to evaluate the effect sizes of colony membership and mother-offspring similarity in both datasets. Specifically, we extracted $R^2$ values for the terms in question after bootstrapping both chemical datasets over individuals. This approach controlled for differences in sample size between the two studies while also providing a visual representation of the magnitude of uncertainty associated with the $R^2$ estimates. We not only found that the patterns reported by *Stoffel et al. (2015)* remained highly significant, but also that the effect size estimates of colony membership and mother-offspring similarity in the two studies were more or less similar, varying by at most a few percent. Elsewhere, in a study that attempted to replicate a hundred psychological studies (*Open Science Collaboration, 2015*), variation in the strength of the original evidence, such as *p*-values, was more predictive of replication success than other characteristics such as the experience or expertise of the original and replication teams. This is consistent with the outcome of the current replication exercise given the high statistical significance ($p < 0.0001$) of the patterns originally reported by *Stoffel et al. (2015)*.

## Generality of the colony membership pattern

We went a step beyond simply repeating our previous study by investigating whether chemical differences between SSB and FWB are specific to these two colonies, or whether chemical signatures are colony-specific in general. Unfortunately, it was not possible to sample mothers from locations other than SSB and FWB due to the difficulty of capturing adult females farther away from the BAS field station where fieldwork on seals is rarely if ever performed. However, the relative ease of capturing pups enabled us to gather a more representative collection of chemical samples from multiple breeding sites around Bird Island. After controlling for the false discovery rate, statistically significant chemical differences were detected in all but two out of 15 pairwise comparisons between colonies. This suggests not only that chemical patterns of colony membership are

repeatable over time, but also that they can be generalized over space. Interestingly, we did not find a clear correspondence between chemical similarity and the geographical proximity of colonies. For example, Freshwater Beach and Main Bay were among the most chemically dissimilar colonies despite being only around 500 m apart, while Johnson Cove and Natural Arch were among the most chemically similar colonies despite being situated at the opposite extremes of Bird Island. The most probable explanation for this pattern is that chemical differences among colonies are predominantly shaped by as yet unknown environmental factors (see below).

## Mechanisms encoding chemical information

Relatively little is currently known about the mechanisms by which colony membership and mother-offspring similarity are chemically encoded in Antarctic fur seals. We know that animals from SSB and FWB exhibit chemical differences despite a lack of genetic differentiation (*Stoffel et al., 2015*), which implies that environmental drivers play an important role. However, it remains unclear exactly what these drivers might be. Food is unlikely to be an important determinant of colony-specific chemical patterns because all of the breeding females around Bird Island feed predominantly on Antarctic krill (*Boyd, Staniland & Martin, 2002*). The underlying substrate is also relatively homogenous, with the vast majority of animals occupying cobblestone breeding beaches that show little in the way of obvious differences to the human eye. It is therefore more likely that colony-specific chemical phenotypes are influenced by differences in local conditions such as temperature, wind or solar radiation, either directly or via alterations to the skin microbiota (*Grosser et al., 2019*). A further possibility could be that chemical differences between colonies reflect differences in microbial communities shaped by social stress. For example, stressful conditions such as high densities of conspecifics can suppress microbial diversity (*Bailey et al., 2011*; *Stothart et al., 2016*; *Noguera et al., 2018*; *Partrick et al., 2018*; *Zha et al., 2018*). This is consistent with our data, as breeding females on SSB are present at higher density and have chronically elevated levels of the stress hormone cortisol (*Meise et al., 2016*), while skin microbial diversity is also lower in this colony (*Grosser et al., 2019*). Investigating the potential linkages between social stress, cortisol, microbial community structure and chemical phenotypes represents a promising avenue for future research.

## CONCLUSIONS

Our study set out to test two hypotheses, namely that chemical patterns of colony membership and mother-offspring similarity in Antarctic fur seals are reproducible over time, and that chemical differences will be present not only between SSB and FWB, but also more generally among colonies. Both hypotheses were supported by our data. The overall robustness of chemical patterns of colony membership and mother-offspring similarity in Antarctic fur seals is consistent with the argument that chemical information is important for social communication in pinnipeds, and lays a solid foundation for future studies of the mechanisms responsible for chemical variation. Finally, as a lack of access to raw data, code and software has been identified as a fundamental obstacle

to replication (*Fidler et al., 2017*), we have made the data from both studies as well as the code used to analyze them freely available, while also using maximally transparent, open access software for peak detection and alignment.

## ACKNOWLEDGEMENTS

We are grateful to the field assistants on Bird Island who contributed toward animal handling and tissue sampling. We would also like to thank John Dickens who helped collect samples from additional beaches on Bird Island. This work contributes to the Ecosystems project of the British Antarctic Survey, Natural Environmental Research Council, and is part of the Polar Science for Planet Earth program.

### Funding

This research was funded by the Deutsche Forschungsgemeinschaft (DFG, German Research Foundation) in the framework of a Sonderforschungsbereich (project numbers 316099922 and 396774617–TRR 212) and the priority program "Antarctic Research with Comparative Investigations in Arctic Ice Areas" (SPP 1158, project number 424119118). It was also supported by core funding from the Natural Environment Research Council to the British Antarctic Survey's Ecosystems Program. Support for the Article Processing Charge was granted by the Deutsche Forschungsgemeinschaft and the Open Access Publication Fund of Bielefeld University. The funders had no role in study design, data collection and analysis, decision to publish, or preparation of the manuscript.

### Grant Disclosures

The following grant information was disclosed by the authors:
Deutsche Forschungsgemeinschaft (DFG, German Research Foundation): 316099922 and 396774617–TRR 212.
Deutsche Forschungsgemeinschaft (DFG, German Research Foundation): Antarctic Research with Comparative Investigations in Arctic Ice Areas: SPP 1158; 424119118.
Natural Environment Research Council Core funding to the British Antarctic Survey's Ecosystems Program.
Open Access Publication Fund of Bielefeld University.

### Competing Interests

The authors declare that they have no competing interests.

### Author Contributions

- Jonas Tebbe conceived and designed the experiments, performed the experiments, analyzed the data, prepared figures and/or tables, authored or reviewed drafts of the paper, and approved the final draft.
- Emily Humble conceived and designed the experiments, performed the experiments, authored or reviewed drafts of the paper, and approved the final draft.

- Martin Adam Stoffel conceived and designed the experiments, performed the experiments, analyzed the data, authored or reviewed drafts of the paper, and approved the final draft.
- Lisa Johanna Tewes performed the experiments, authored or reviewed drafts of the paper, and approved the final draft.
- Caroline Müller conceived and designed the experiments, analyzed the data, authored or reviewed drafts of the paper, and approved the final draft.
- Jaume Forcada conceived and designed the experiments, authored or reviewed drafts of the paper, and approved the final draft.
- Barbara Caspers conceived and designed the experiments, analyzed the data, authored or reviewed drafts of the paper, and approved the final draft.
- Joseph Ivan Hoffman conceived and designed the experiments, analyzed the data, prepared figures and/or tables, authored or reviewed drafts of the paper, and approved the final draft.

## Animal Ethics

The following information was supplied relating to ethical approvals (i.e., approving body and any reference numbers):

All field procedures were approved by the British Antarctic Survey Animal Welfare and Ethics Review Body (reference no. PEA6).

## Field Study Permissions

The following information was supplied relating to field study approvals (i.e., approving body and any reference numbers):

Samples were collected as part of the Polar Science for Planet Earth program of the British Antarctic Survey under the authorization of the Senior Executive and the Environment Officers of the Government of South Georgia and the South Sandwich Islands (permit no. 2016/013).

## Data Availability

The code is available in the Supplemental Files. All files (code, code snippets and used data) used for analysis are available at GitHub: https://github.com/tebbej/SealScent2020. *Stoffel et al. (2015)* data are available from GitHub: https://github.com/mastoffel/seal_chemical_fingerprints.

## Supplemental Information

Supplemental information for this article can be found online at http://dx.doi.org/10.7717/peerj.10131#supplemental-information.

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
