# Peer review of "Chemical patterns of colony membership and mother-offspring similarity in Antarctic fur seals are reproducible"

_PeerJ, doi:10.7717/peerj.10131_

## Round 0.1 · original submission · Major Revisions

Dear author,
Both referees recognized the interest of this study and made many interesting comments to improve the flow and clarity of your manuscript.
One referee has however two more serious concerns, one about statistics, and one more general about the mixing between repetition (aim of the manuscript)on the two colonies and the extention to six others non precedenlty sampled.
I am also concerned by this mixture of results especially because I am puzzled by the fact that 1) odors seem to be used for mother-offsrping recognition and 2) that you use colonies differences to emphasize this possibility. are mothers often changing of colonies that colonies differences help recognizing her offspring? I just do not understand the reasoning here. This apparent confusion (to my eyes) might be removed if you dollow the suggestion by the second referee to drop the results concerning the six additional colonies.
This is why I recommend major revisions and expect that you will be able to clarify these points
Best wishes for the revision

Reviewer 1 ·

Basic reporting

The manuscript is exceptionally well written and appropriately structured, making it very easy to read and understand. The text flows well and the concepts are thoroughly explained. The knowledge of the authors on the background of the study is evident and the references are appropriate and up to date.

Below I have listed a few minor comments about the background and the general outline of the publication that I hope the authors will find the following useful in improving their manuscript:

At the moment, the introduction and therefore the framing of the whole article is focused on replication.
1) Please rephrase the title to reflect this more. After reading the current title, I expected a very different introduction to the article, which was slightly confusing at the beginning.
2) I fully support replication studies and consider them an extremely important part of research, as outlined in the introduction. However, the article seems to have two separate goals: a) replicating a subsection of results presented in Stoffel et al. 2015 and b) adding additional information about chemical profile differences among colonies. Adding original results to the study (and their discussion) causes the replication section to lose importance, as it creates the feeling of a side research project that was difficult to publish due to the lack of novelty and was then restructured into a study highlighting its replication aspect. I realise the difficulties of publishing replicated studies and wanting to publish all available results, however I think the paper would benefit if just one, clear aim was presented – the replication study. The analysis is well executed, well explained and provides enough content for a publication and can be used as guide for future studies of this sort. The additional results, presenting chemical profile differences among multiple colonies, are interesting and worth publishing. However, I think it would be better to publish these as a standalone short article or note.

L54-62: Rather than providing quotes from the referenced papers, please summarise and rephrase the content, leaving just the reference at the end of the sentence.

The current introduction contains 5 paragraphs on replication. While most information is relevant to the current study and revisited in the discussion, paragraphs 4 and 5 could be made more concise.

L101: Please remove the abbreviation - it is not used later

L133-140: Please also mention the inconsistencies/lack of appropriate reporting/lack of automated (and therefore reproducible) peak detection and alignment methods. This creates many inconsistencies and difficulties in replicating results using GCMS data.

Experimental design

The experimental design followed the excellent study of Stoffel et al. 2015 and have no major concerns regarding it. The authors also clearly pointed out how it differed from the original paper and how that could have affected their results, which I greatly appreciated.
The code and raw data from the original as well as the replicated study are made available making it transparent and reproducible.

Minor comments:
L192-193: Were mother-pup pairs caught at the same time/on the same day/when together? If they were, the mother-offspring similarity found in the study could be a result of animal touching and transferring compounds from one another directly prior to capture and may not be a reflection of ‘stable’ similarity.

L228: Why was the PERMANOVA based on 99999 permutations? This seems like a lot. The default and what is used in most publications is a lot less. Did lower permutations result in insignificant values?

Table 1c: It seems that you re-analysed the raw data of Stoffel et al. 2015 using a PERMANOVA. Please add this information to the text. Did you also realign the raw data using GCalignR (to keep it consistent with your analysis)?

L237: It is not clear what you mean by ‘we did not re-align the dataframe of Stoffel et al.’. Do you mean that you did not use GCalignR and kept the original alignment? If so, please make this clear (as it is not stated that you realigned the data from the original paper for the PERMANOVA in Table 1c (see above)).

Supplementary materials: the code that was provided imports data with peaks that are already aligned. It would be useful to provide the code for aligning peaks in the supplementary materials (rather than just the github repository) or briefly describing how this was done in the text.

Validity of the findings

As mentioned before, I agree with the authors that replications studies are important and needed. In the field of ecology, this may be particularly difficult, and I commend the authors’ effort of collecting an impressive dataset and conducting this analysis again. The rationale and benefit of this research is clearly stated and the methods are well developed and sound. The discussion is well explained and flows well.

I have only one major concern with the interpretation of the results – the significant result of the test for homogeneity of group variances (L274-277). This result could mean that the significant results of the PERMANOVA are actually an artefact of differences in within-group variability, rather than between group differences, which is a problem and theoretically puts the entire results into question. In lines 267-278 you mention a post hoc test – this is not included in the code in the supplementary materials. Considering these are the main results of the article, I would like to see more justification and support of why you think significant results of the betadisper analysis do not indicate issues and potential unreliability of the results of the PERMANOVA as well as explanation for considering the post hoc test results more reliable than the original test.

A few minor comments:
Since you included age in your model, you may consider running betadisper for scent_factor$age

Figure 2B – There are a lot of colours and shapes making it difficult to find pairs. Please make this plot clearer, maybe by adding lines connecting mother-pup pairs. You could then merge 2A and 2B into one plot, by keeping the colour and shapes of 2A and connecting mother-pup pairs with lines.

L291-296: As mentioned above, I think the replication study is enough for this publication and this section could be left out.

L382-387: This statement is too much of a reach. Just because mother-offspring similarity and colony differences persist does not mean that this is related by microbiota. While this may be true and your findings do not contradict this possibility, what you have shown in this study does not relate to this enough to bring this up, please remove.

L402-403. There are a lot of other factors that may play a role in this and there is no need to make broad speculations. Especially since the main goal of the publication was to replicate a study. Please remove.

L405-425: This was not within the scope of the study and I do not see the need to discuss it here. Considering that the main body and framework of the study focuses on replication of results, the whole section on mechanisms is more appropriate in the discussion of the original paper.

·

Basic reporting

The manuscript is very well written and a pleasure to read. The authors have provided ample background, both of the issue of replication and of pinniped olfactory signatures. The article is well structured and the methods are clear. The results relate directly to the hypotheses and the conclusions are well supported.

Specific comments:
The reference to Ling on line 146 appears to have a typo in it.
In the reference list, some of the binomial names are not italicised.

Experimental design

The research question is well defined and very relevant. Chemical ecology is a developing field with many different methods being trialled by different labs and the reliability and comparability of those results have not yet been established. This study is an important step in showing that, at least using the methods of Stoffel et al., findings are robust and reproducible.

One additional detail I would request in the methods is the duration of time from sample collection to freezing, and the length of time samples were stored before analysis. This is not a critique of this study, but a detail that I think is useful to other researchers as the stability of chemical samples post collection is poorly understood and providing this information establishes a benchmark.

Validity of the findings

The authors have taken great care with the statistics in this study. The results are clearly presented and the conclusions are supported by the results.

Additional comments

Thank you for this very interesting paper. It provides both important findings about pinniped chemical communication, and about methodological approaches.

I find it interesting that Johnsons Cove and Natural Arch were similar to each other (as they are the furthest apart), while LB, SSB and FWB were all different yet close to each other. It is probably beyond the speculation of this paper, but it will be interesting to see more investigation in the future into the drivers of similarity (or diversity) between colonies and individuals.

Given this study was a replication of Stoffel et al, it isn't appropriate to introduce new methods, but with the potential environmental influence in mind, have you considered testing environmental control samples? Wierucka et al., used environmental controls in an attempt to remove some of the contribution of other odour sources in the colony. Given these methods are relatively new, it would be good to determine if such controls are necessary, and/or if they contribute to our understanding of the sources of diversity among colonies. Again, this is probably beyond this paper, but a potential avenue of future research.

---

## Round 0.2 · accepted · Accept

Although I am still not entirely convinced by the arguments for extension replacing almost true replication, I think the results and story are interesting enough and consistent with our readership. My advice is to accept your corrected manuscript.

Reviewer 1 ·

Basic reporting

No comment

Experimental design

No comment

Validity of the findings

No comment

Additional comments

Thank you very much for this new and improved version of the manuscript. I am happy with the changes made with regard to my comments about the experimental design and validity of findings. I especially appreciate the additional analyses conducted to explore certain topics in depth. I believe that the reworked sections of the introduction make it clearer why the two parts of the paper were combined.